# Deep Bidirectional Language-Knowledge Graph Pretraining

**Michihiro Yasunaga,**[1]  **Antoine Bosselut,**[2]  **Hongyu Ren,**[1]  **Xikun Zhang**[1]
**Christopher D Manning,**[1]  **Percy Liang,**[1*]  **Jure Leskovec**[1*]
[1]Stanford University  [2]EPFL  [*]Equal senior authorship
{myasu,antoineb,hyren,xikunz2,manning,pliang,jure}@cs.stanford.edu

## Abstract

Pretraining a language model (LM) on text has been shown to help various downstream NLP tasks. Recent works show that a knowledge graph (KG) can complement text data, offering structured background knowledge that provides a useful scaffold for reasoning. However, these works are not pretrained to learn a deep fusion of the two modalities at scale, limiting the potential to acquire fully joint representations of text and KG. Here we propose DRAGON (**D**eep Bidi**r**ectional **La**ngu**a**ge-Kn**o**wledge Graph Pretrai**n**ing), a self-supervised method to pretrain a deeply joint language-knowledge foundation model from text and KG at scale. Specifically, our model takes pairs of text segments and relevant KG subgraphs as input and bidirectionally fuses information from both modalities. We pretrain this model by unifying two self-supervised reasoning tasks, masked language modeling and KG link prediction. DRAGON outperforms existing LM and LM+KG models on diverse downstream tasks including question answering across general and biomedical domains, with +5% absolute gain on average. In particular, DRAGON achieves strong performance on complex reasoning about language and knowledge (+10% on questions involving long contexts or multi-step reasoning) and low-resource QA (+8% on OBQA and RiddleSense), and new state-of-the-art results on various BioNLP tasks. Our code and trained models are available at https://github.com/michiyasunaga/dragon.

## 1  Introduction

Pretraining learns self-supervised representations from massive raw data to help various downstream tasks [1]. Language models (LMs) pretrained on large amounts of text data, such as BERT [2] and GPTs [3], have shown strong performance on many natural language processing (NLP) tasks. The success of these models comes from deeply interactive (contextualized) representations of input tokens learned at scale via self-supervision [2, 4]. Meanwhile, large knowledge graphs (KGs), such as Freebase [5], Wikidata [6] and ConceptNet [7], can provide complementary information to text data. KGs offer structured background knowledge by representing entities as nodes and relations between them as edges, and also offer scaffolds for structured, multi-step reasoning about entities [8, 9, 10, 11] (§3.4.1). The dual strengths of text data and KGs motivate research in pretraining deeply interactive representations of the two modalities at scale.

How to effectively combine text and KGs for pretraining is an open problem and presents challenges. Given text and KG, we need both (i) a *deeply bidirectional* model for the two modalities to interact, and (ii) a *self-supervised* objective to learn joint reasoning over text and KG at scale. Several existing works [12, 13, 14, 15, 16] propose methods for self-supervised pretraining, but they fuse text and KG in a shallow or uni-directional manner. Another line of work [8, 9] proposes bidirectional models for text and KG, but these models focus on finetuning on labeled downstream tasks and do not perform

36th Conference on Neural Information Processing Systems (NeurIPS 2022).

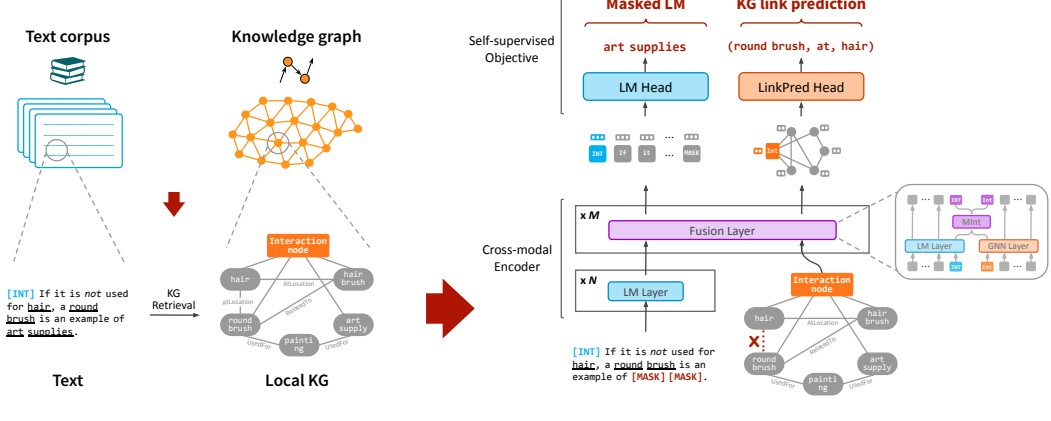

Figure 1: **Overview of our approach, DRAGON.** *Left*: Given raw data of a text corpus and a large knowledge graph, we create aligned (text, local KG) pairs by sampling a text segment from the corpus and extracting a relevant subgraph from the KG (§2.1). As the structured knowledge in KG can ground the text and the text can provide the KG with rich context for reasoning, we aim to pretrain a language-knowledge model jointly from the text-KG pairs (DRAGON). *Right*: To model the interactions over text and KG, DRAGON uses a cross-modal encoder that bidirectionally exchanges information between them to produce fused text token and KG node representations (§2.2). To pretrain DRAGON jointly on text and KG, we unify two self-supervised reasoning tasks: (1) masked language modeling, which masks some tokens in the input text and then predicts them, and (2) link prediction, which holds out some edges from the input KG and then predicts them. This joint objective encourages text and KG to mutually inform each other, facilitating the model to learn joint reasoning over text and KG (§2.3).

self-supervised learning. Consequently, existing methods may have limited their potential to model and learn deep interactions over text and KG.

To address both of the above challenges and fully unify the strengths of text and KG, we propose **DRAGON** (**D**eep Bidi**r**ectional L**a**n**g**uage-Kn**o**wledge Graph Pretrai**n**ing), an approach that performs deeply bidirectional, self-supervised pretraining of a language-knowledge model from text and KG. DRAGON has two core components: a cross-modal model that bidirectionally fuses text and KG, and a bidirectional self-supervised objective that learns joint reasoning over text and KG. Concretely, as in Figure 1, we take a text corpus and a KG as raw data, and create inputs for the model by sampling a text segment from the corpus and extracting a relevant subgraph from the KG via entity linking, obtaining a (*text*, *local KG*) pair. We use a cross-modal model to encode this input into fused representations, where each layer of the model encodes the text with an LM and the KG with a graph neural network (GNN), and fuses the two with a bidirectional modality interaction module (GreaseLM [9]). We pretrain this model by unifying two self-supervised reasoning tasks: (1) masked language modeling (MLM), which masks and predicts tokens in the input text, and (2) link prediction, which drops and predicts edges in the input KG. The intuition is that by combining the two tasks, MLM makes the model use the text jointly with structured knowledge in the KG to reason about masked tokens in the text (e.g., in Figure 1, using the "round brush"–"art supply" multi-hop path from the KG helps), and link prediction makes the model use the KG structure jointly with the textual context to reason about missing links in the KG (e.g., recognizing that "round brush could be used for hair" from the text helps). This joint objective thus enables text to be grounded by KG structure and KG to be contextualized by text simultaneously, producing a deeply-unified language-knowledge pretrained model where information flows bidirectionally between text and KG for reasoning.

We pretrain DRAGON in two domains: a general domain, using the Book corpus and ConceptNet KG [7] (§3), and a biomedical domain, using the PubMed corpus and UMLS KG [17] (§4). We show that DRAGON improves on existing LM and LM+KG models on diverse downstream tasks across domains. For the general domain, DRAGON outperforms RoBERTa [18], our base LM without KGs, on various commonsense reasoning tasks such as CSQA, OBQA, RiddleSense and HellaSwag, with +8% absolute accuracy gain on average. For the biomedical domain, DRAGON improves on the previous best LM, BioLinkBERT [19], and sets a new state of the art on BioNLP tasks such as MedQA and PubMedQA, with +3% accuracy gain. In particular, DRAGON exhibits notable improvements on QA tasks involving complex reasoning (+10% gain on multi-step, negation, hedge, or long context reasoning) and on downstream tasks with limited training data (+8% gain). These results show that our deep bidirectional self-supervision over text and KG produces significantly improved language-knowledge representations compared to existing models.

## 1.1 Related work

**Knowledge-augmented LM pretraining.** Knowledge integration is active research for improving LMs. One line of works is retrieval-augmented LMs [20, 21, 22], which retrieve relevant text from a corpus and integrate it into LMs as additional knowledge. Orthogonal to these works, we focus on using knowledge bases as background knowledge, to ground reasoning about entities and facts.

Closest to our work are works that integrate knowledge bases in LM pretraining. One line of research aims to add entity features to LMs [12, 23, 24]; Some works use the KG entity information or structure to create additional training signals [13, 25, 14, 26, 27, 28]; Several works add KG triplet information directly to the LM input [29, 16, 15, 30, 31]. While these methods have achieved substantial progress, they typically propagate information between text and KG in a shallow or uni-directional (e.g., KG to text) manner, which might limit the potential to perform fully joint reasoning over the two modalities. To improve on the above works, we propose to bidirectionally interact text and KG via a deep cross-modal model and joint self-supervision, so that text and KG are grounded and contextualized by each other. We find that this improves model performance on various reasoning tasks (§3). Another distinction is that existing works in this space typically focus on adding entity- or triplet-level knowledge from KGs to LMs, and focus on solving entity/relation classification tasks. Our work significantly expands this scope in that we use larger KG subgraphs (200 nodes) as input to enable richer contextualization between KG and text, and we achieve performance improvements on a broader set of NLP tasks including QA, reasoning and text classification tasks.

**KG-augmented question answering.** Various works designed KG-augmented reasoning models for question answering [32, 33, 34, 35, 36, 37, 38, 39, 40, 41, 42]. In particular, recent works such as QA-GNN [8] and GreaseLM [9] suggest that a KG can scaffold reasoning about entities with its graph structure, and help for complex question answering (e.g., negation, multi-hop reasoning). These works typically focus on training or finetuning models on particular QA datasets. In contrast, we generalize this and integrate KG-augmented reasoning into general-purpose pretraining. Our motivation is that self-supervised pretraining allows the model to learn from larger and more diverse data, helping to learn richer interactions between text and KGs and to acquire more diverse reasoning abilities beyond specific QA tasks. We find that our proposed pretraining approach (DRAGON) offers significant boosts over the baseline QA models (e.g. GreaseLM) on diverse downstream tasks (§3). This opens a new research avenue in scaling up various carefully-designed QA models to pretraining.

**KG representation learning.** Our link prediction task used in pretraining is motivated by research in KG representation learning. Link prediction is a fundamental task in KGs [43, 44], and various works study methods to learn KG entity and relation embeddings for link prediction, such as TransE [45], DistMult [46] and RotatE [47]. Several works additionally use textual data or pretrained LMs to help learn KG embeddings and link prediction [48, 49, 50, 51, 52, 53]. While these works focus on the KG-side representations, we extend the scope and use the KG-side objective (link prediction) jointly with a text-side objective (language modeling) to train a mutually-interactive text-KG model.

## 2 Deep Bidirectional Language-Knowledge Graph Pretraining (DRAGON)

We propose DRAGON, an approach that performs deeply bidirectional, self-supervised pretraining of a language-knowledge model from text and KG. Specifically, as illustrated in Figure 1, we take a text corpus and a large knowledge graph as raw data, and create input instances for the model by sampling coarsely-aligned (text segment, local KG) pairs (§2.1). To learn mutual interactions over text and KG, DRAGON consists of a cross-modal encoder (GreaseLM) that fuses the input text-KG pair bidirectionally (§2.2), and a pretraining objective that performs bidirectional self-supervision on the text-KG input (§2.3). Our pretraining objective unifies masked language modeling (MLM) and KG link prediction (LinkPred) to make text and KG mutually inform each other and learn joint reasoning over them. Finally, we describe how we finetune the pretrained DRAGON model for downstream tasks (§2.4). While each individual piece of our approach (GreaseLM, MLM, LinkPred) is not new in itself, we are the first to bring them together effectively and demonstrate that the resulting model has strong empirical results (§3, §4).

**Definitions.** We define a text corpus $\mathcal{W}$ as a set of text segments $\mathcal{W} = \{W\}$, and each text segment $W$ as a sequence of tokens (words), $W = (w_1, ..., w_I)$. We define a knowledge graph (KG) as a multi-relational graph $\mathcal{G} = (\mathcal{V}, \mathcal{E})$, where $\mathcal{V}$ is the set of entity nodes in the KG and $\mathcal{E} \subseteq \mathcal{V} \times \mathcal{R} \times \mathcal{V}$ is the set of edges (triplets) that connect nodes in $\mathcal{V}$, with $\mathcal{R}$ being the set of relation types $\{r\}$.

Each triplet $(h, r, t)$ in a KG can represent a knowledge fact such as $(\texttt{Paris}, \texttt{in}, \texttt{France})$. As a raw KG is often large, with millions of nodes, a subgraph of the raw KG (*local KG*) is considered: $G = (V, E)$ where $V = \{v_1, ..., v_J\} \subseteq \mathcal{V}$ and $E \subseteq \mathcal{E}$. We define a language-knowledge model to be a composition of two functions, $f_{\text{head}}(f_{\text{enc}}(X))$, where the encoder $f_{\text{enc}}$ takes in an input $X = $ (text segment $W$, local KG $G$), and produces a contextualized vector representation for each text token, $(\mathbf{H}_1, ..., \mathbf{H}_I)$, and for each KG node, $(\mathbf{V}_1, ..., \mathbf{V}_J)$. A language model is a special case of a language-knowledge model with no KG ($J = 0$). The head $f_{\text{head}}$ uses these representations to perform self-supervised tasks in the pretraining step and to perform downstream tasks in the finetuning step.

## 2.1 Input representation

Given a text corpus $\mathcal{W}$ and a large knowledge graph $\mathcal{G}$, we create input instances for the model by preparing (text segment $W$, local KG $G$) pairs. We want each pair's text and KG to be (roughly) semantically aligned so that the text and KG can mutually inform each other and facilitate the model to learn interactive reasoning between the two modalities. Specifically, for each text segment $W$ from $\mathcal{W}$, we extract a relevant local KG $G$ for it from $\mathcal{G}$ via the following KG retrieval process.

**KG retrieval.** Given a text segment $W$, we link entity mentions in $W$ to entity nodes in $\mathcal{G}$ to get an initial set of nodes $V_{\text{el}}$. We then add their 2-hop bridge nodes from $\mathcal{G}$ to get the total retrieved nodes $V \subseteq \mathcal{V}$. Lastly, we add all edges that span these nodes in $\mathcal{G}$ to get $E \subseteq \mathcal{E}$, which yields the final local KG, $G = (V, E)$, as well as our final input instance $X = (W, G)$. Appendix B.1 provides more details on KG retrieval. Henceforth, we use "KG" to refer to this local KG $G$ unless noted otherwise.

**Modality interaction token/node.** For each resulting (text, KG) pair, we further add a special token (interaction token) $w_{\text{int}}$ to the text and a special node (interaction node) $v_{\text{int}}$ to the KG, which will serve as an information pooling point for each modality as well as an interface for modality interaction in our cross-modal encoder (§2.2). Specifically, we prepend $w_{\text{int}}$ to the original text $W = (w_1, ..., w_I)$, and connect $v_{\text{int}}$ to the entity-linked nodes in the original KG, $V_{\text{el}} \subseteq V = \{v_1, ..., v_J\}$, using a new relation type $r_{\text{el}}$. The interaction token and node can also be used to produce a pooled representation of the whole input, e.g., when finetuning for classification tasks (§2.4).

## 2.2 Cross-modal encoder

To model mutual interactions over the text and KG, we use a bidirectional sequence-graph encoder for $f_{\text{enc}}$ which takes in the text tokens and KG nodes and exchanges information across them for multiple layers to produce a fused representation of each token and node (Figure 1 right):

$$(\mathbf{H}_{\text{int}}, \mathbf{H}_1, ..., \mathbf{H}_I), (\mathbf{V}_{\text{int}}, \mathbf{V}_1, ..., \mathbf{V}_J) = f_{\text{enc}}((w_{\text{int}}, w_1, ..., w_I), (v_{\text{int}}, v_1, ..., v_J)) \quad (1)$$

While we may use any deep bidirectional sequence-graph encoder for $f_{\text{enc}}$, for controlled comparison with existing works, we adopt the existing top-performing sequence-graph architecture, GreaseLM [9], which combines Transformers [54] and graph neural networks (GNNs) to fuse text-KG inputs.

Specifically, GreaseLM first uses $N$ layers of Transformer language model (LM) layers to map the input text into initial token representations, and uses KG node embeddings to map the input KG nodes into initial node representations,

$$(\mathbf{H}_{\text{int}}^{(0)}, \mathbf{H}_1^{(0)}, ..., \mathbf{H}_I^{(0)}) = \text{LM-Layers}(w_{\text{int}}, w_1 ..., w_I), \quad (2)$$

$$(\mathbf{V}_{\text{int}}^{(0)}, \mathbf{V}_1^{(0)}, ..., \mathbf{V}_J^{(0)}) = \text{Node-Embedding}(v_{\text{int}}, v_1, ..., v_J). \quad (3)$$

Then it uses $M$ layers of text-KG fusion layers to encode these token/node representations jointly into the final token/node representations,

$$(\mathbf{H}_{\text{int}}, ..., \mathbf{H}_I), (\mathbf{V}_{\text{int}}, ..., \mathbf{V}_J) = \text{Fusion-Layers}((\mathbf{H}_{\text{int}}^{(0)}, ..., \mathbf{H}_I^{(0)}), (\mathbf{V}_{\text{int}}^{(0)}, ..., \mathbf{V}_J^{(0)})), \quad (4)$$

where each of the fusion layers ($\ell = 1, ..., M$) performs the following:

$$(\widetilde{\mathbf{H}}_{\text{int}}^{(\ell)}, \mathbf{H}_1^{(\ell)}, ..., \mathbf{H}_I^{(\ell)}) = \text{LM-Layer}(\mathbf{H}_{\text{int}}^{(\ell-1)}, \mathbf{H}_1^{(\ell-1)}, ..., \mathbf{H}_I^{(\ell-1)}), \quad (5)$$

$$(\widetilde{\mathbf{V}}_{\text{int}}^{(\ell)}, \mathbf{V}_1^{(\ell)}, ..., \mathbf{V}_J^{(\ell)}) = \text{GNN-Layer}(\mathbf{V}_{\text{int}}^{(\ell-1)}, \mathbf{V}_1^{(\ell-1)}, ..., \mathbf{V}_J^{(\ell-1)}), \quad (6)$$

$$[\mathbf{H}_{\text{int}}^{(\ell)}; \mathbf{V}_{\text{int}}^{(\ell)}] = \text{MInt}([\widetilde{\mathbf{H}}_{\text{int}}^{(\ell)}; \widetilde{\mathbf{V}}_{\text{int}}^{(\ell)}]). \quad (7)$$

Here GNN induces graph structure-aware representations of KG nodes, $[\cdot ; \cdot]$ does concatenation, and MInt (modality interaction module) exchanges information between the interaction token (text side) and interaction node (KG side) via an MLP. For more details on GreaseLM, we refer readers to [9].

## 2.3 Pretraining objective

We aim to pretrain the DRAGON model so that it learns joint reasoning over text and a KG. To ensure that the text and KG mutually inform each other and the model learns bidirectional information flow, we unify two self-supervised reasoning tasks: masked language modeling and KG link prediction.

**Masked language modeling (MLM).** MLM is a common pretraining task used for language models (e.g., BERT [2], RoBERTa [18]), which masks some tokens in the input text and predicts them. This task makes the model use non-masked context to reason about masked tokens, and in particular, as our approach takes a joint text-KG pair as input, we expect that MLM can encourage the model to learn to use the text *jointly with* structured knowledge in the KG to reason about masks in the text (e.g., in the example of Figure 1, besides the textual context, recognizing the "round brush"–"art supply" path from the KG can help together to predict the masked tokens "art supplies").

Concretely, to perform the MLM task, we mask a subset of tokens in the input text, $M \subseteq W$, with a special token [MASK], and let the task head $f_{\text{head}}$ be a linear layer that takes the contextualized token vectors $\{\mathbf{H}_i\}$ from the encoder to predict the original tokens. The objective is a cross-entropy loss:

$$\mathcal{L}_{\text{MLM}} = -\sum_{i \in M} \log p(w_i \mid \mathbf{H}_i). \tag{8}$$

**Link prediction (LinkPred).** While the MLM task predicts for the text side, link prediction holds out some edges and predicts them for the input KG. Link prediction is a fundamental task in KGs [47] and makes the model use the structure of KGs to perform reasoning (e.g., using a compositional path "X's mother's husband is Y" to deduce a missing link "X's father is Y"). In particular, as our approach takes a joint text-KG pair as input, we expect that link prediction can encourage the model to learn to use the KG structure *jointly with* the textual context to reason about missing links in the KG (e.g., in Figure 1, besides the KG structure, recognizing that "round brush could be used for hair" from the text can help together to predict the held-out edge (round_brush, at, hair)).

Concretely, to perform the link prediction task, we hold out a subset of edge triplets from the input KG, $S = \{(h, r, t)\} \subseteq E$. For the task head $f_{\text{head}}$, we adopt a KG representation learning framework, which maps each entity node ($h$ or $t$) and relation ($r$) in the KG to a vector, $\mathbf{h}, \mathbf{t}, \mathbf{r}$, and defines a scoring function $\phi_r(\mathbf{h}, \mathbf{t})$ to model positive/negative triplets. Specifically, we let $\mathbf{h} = \mathbf{V}_h$, $\mathbf{t} = \mathbf{V}_t$, $\mathbf{r} = \mathbf{R}_r$, with $\{\mathbf{V}_j\}$ being the contextualized node vectors from the encoder, and $\mathbf{R} = \{\mathbf{r}_1, ..., \mathbf{r}_{|\mathcal{R}|}\}$ being learnable relation embeddings. We consider a KG triplet scoring function $\phi_r(\mathbf{h}, \mathbf{t})$ such as

$$\text{DistMult [46]: } \langle \mathbf{h}, \mathbf{r}, \mathbf{t} \rangle, \quad \text{TransE [45]: } -\|\mathbf{h} + \mathbf{r} - \mathbf{t}\|, \quad \text{RotatE [47]: } -\|\mathbf{h} \odot \mathbf{r} - \mathbf{t}\|, \tag{9}$$

where $\langle \cdot, \cdot, \cdot \rangle$ denotes the trilinear dot product and $\odot$ the Hadamard product. A higher $\phi$ indicates a higher chance of $(h, r, t)$ being a positive triplet (edge) instead of negative (no edge). We analyze the choices of scoring functions in §3.4.3. For training, we optimize the objective:

$$\mathcal{L}_{\text{LinkPred}} = \sum_{(h,r,t) \in S} \left( -\log \sigma(\phi_r(\mathbf{h}, \mathbf{t}) + \gamma) + \frac{1}{n} \sum_{(h',r,t')} \log \sigma(\phi_r(\mathbf{h}', \mathbf{t}') + \gamma) \right), \tag{10}$$

where $(h', r, t')$ are $n$ negative samples corresponding to the positive triplet $(h, r, t)$, $\gamma$ is the margin, and $\sigma$ is the sigmoid function. The intuition of this objective is to make the model predict triplets of the held-out edges $S$ as positive and other random triplets as negative.

**Joint training.** To pretrain DRAGON, we optimize the MLM and LinkPred objectives jointly: $\mathcal{L} = \mathcal{L}_{\text{MLM}} + \mathcal{L}_{\text{LinkPred}}$. This joint objective unifies the effects of MLM and LinkPred, which encourage the model to simultaneously ground text with KG structure and contextualize KG with text, facilitating bidirectional information flow between text and KGs for reasoning. We show in §3.4.3 that the joint objective yields a more performant model than using one of the objectives alone.

## 2.4 Finetuning

Lastly, we describe how we finetune DRAGON for downstream tasks such as text classification and multiple-choice QA (MCQA). Given an input text $W$ (e.g., concatenation of a question and an answer choice in the case of MCQA), we follow the same steps as §2.1 and §2.2 to retrieve a relevant local KG $G$ and encode them jointly into contextualized token/node vectors, $(\mathbf{H}_{\text{int}}, \mathbf{H}_1, ...,$

$\mathbf{H}_I$), $(\mathbf{V}_{\text{int}}, \mathbf{V}_1, ..., \mathbf{V}_J)$. We then compute a pooled representation of the whole input as $\mathbf{X} = \text{MLP}(\mathbf{H}_{\text{int}}, \mathbf{V}_{\text{int}}, \mathbf{G})$, where $\mathbf{G}$ denotes attention-based pooling of $\{\mathbf{V}_j \mid v_j \in \{v_1, ..., v_J\}\}$ using $\mathbf{H}_{\text{int}}$ as a query. Finally, the pooled representation $\mathbf{X}$ is used to perform the downstream task, in the same way as how the `[CLS]` representation is used in LMs such as BERT and RoBERTa.

The difference from GreaseLM is that while GreaseLM only performs finetuning as described in this section (hence, it is an LM *finetuned* with KGs), DRAGON performs self-supervised pretraining as described in §2.3 (hence, it can be viewed as an LM *pretrained + finetuned* with KGs).

## 3   Experiments: General domain

We experiment with the proposed approach DRAGON in a general domain first. We pretrain DRAGON using the Book corpus and ConceptNet KG (§3.1), and evaluate on diverse downstream tasks (§3.2). We show that DRAGON significantly improves on existing models (§3.4). We extensively analyze the effect of DRAGON's key design choices such as self-supervision and use of KGs (§3.4.1, 3.4.2, 3.4.3). We also experiment in the biomedical domain in §4.

### 3.1   Pretraining setup

**Data.**   For the text data, we use documents involving commonsense, BookCorpus [55]. BookCorpus has 6GB of text from online books and is widely used in LM pretraining (e.g., BERT, RoBERTa). For the KG data, we use ConceptNet [7], a general-domain knowledge graph designed to capture background commonsense knowledge. It has 800K nodes and 2M edges in total. To create a training instance, we sample a text segment of length up to 512 tokens from the text corpus, then retrieve a relevant KG subgraph of size up to 200 nodes (details in Appendix B.1), by which we obtain an aligned (text, local KG) pair.

**Implementation.**   For our encoder (§2.2), we use the exact same architecture as GreaseLM [9] (19 LM layers followed by 5 text-KG fusion layers; 360M parameters in total). As done by [9], we initialize parameters in the LM component with the RoBERTa-Large release [18] and initialize the KG node embeddings with pre-computed ConceptNet entity embeddings (details in Appendix B.2). For the link prediction objective (§2.3, Equation 10), we use DistMult [46] for KG triplet scoring, with a negative exampling of 128 triplets and a margin of $\gamma = 0$. To pretrain the model, we perform MLM with a token masking rate of 15% and link prediction with an edge drop rate of 15%. We pretrain for 20,000 steps with a batch size of 8,192 and a learning rate of 2e-5 for parameters in the LM component and 3e-4 for the others. Training took 7 days on eight A100 GPUs using FP16. Additional details on the hyperparameters can be found in Appendix B.3.

### 3.2   Downstream evaluation tasks

We finetune and evaluate DRAGON on nine diverse commonsense reasoning benchmarks: Common-senseQA (**CSQA**) [56], OpenbookQA (**OBQA**) [57], RiddleSense (**Riddle**) [58], AI2 Reasoning Challenge−Challenge Set (**ARC**) [59], **CosmosQA** [60], **HellaSwag** [61], Physical Interaction QA (**PIQA**) [62], Social Interaction QA (**SIQA**) [63], and Abductive Natural Language Inference (**aNLI**) [64]. For CSQA, we follow the in-house data splits used by prior works [32]. For OBQA, we follow the original setting where the models only use the question as input and do not use the extra science facts. Appendix B.4 provides the full details on these tasks and data splits. Hyperparameters used for finetuning can be found in Appendix B.3.

### 3.3   Baselines

**LM.**   To study the effect of using KGs, we compare DRAGON with the vanilla language model, RoBERTa [18]. As we initialize DRAGON's parameters using the RoBERTa-Large release (§3.1), for fair comparison, we let the baseline be such that we take the RoBERTa-Large release and continue pretraining it with the vanilla MLM objective on the same text data for the same number of steps as DRAGON. Hence, the only difference is that DRAGON uses KGs during pretraining while RoBERTa does not. We then perform standard LM finetuning of RoBERTa on downstream tasks.

**LM finetuned with KG.**   We also compare with existing KG-augmented QA models, QAGNN [8] and GreaseLM [9], which *finetune* a vanilla LM (i.e. RoBERTa-Large) with a KG on downstream

|  | CSQA | OBQA | Riddle | ARC | CosmosQA | HellaSwag | PIQA | SIQA | aNLI |
|---|---|---|---|---|---|---|---|---|---|
| RoBERTa [18] | 68.7 | 64.9 | 60.7 | 43.0 | 80.5 | 82.3 | 79.4 | 75.9 | 82.7 |
| QAGNN [8] | 73.4 | 67.8 | 67.0 | 44.4 | 80.7 | 82.6 | 79.6 | 75.7 | 83.0 |
| GreaseLM [9] | 74.2 | 66.9 | 67.2 | 44.7 | 80.6 | 82.8 | 79.6 | 75.5 | 83.3 |
| DRAGON (**Ours**) | **76.0** | **72.0** | **71.3** | **48.6** | **82.3** | **85.2** | **81.1** | **76.8** | **84.0** |

Table 1: Accuracy on downstream commonsense reasoning tasks. DRAGON consistently outperforms the existing LM (RoBERTa) and KG-augmented QA models (QAGNN, GreaseLM) on all tasks. The gain is especially significant on tasks that have small training data (*OBQA*, *Riddle*, *ARC*) and tasks that require complex reasoning (*CosmosQA*, *HellaSwag*).

|  | Negation | Conjunction | Hedge | # Prepositional Phrases | | | | # Entities |
|---|---|---|---|---|---|---|---|---|
|  |  |  |  | 0 | 1 | 2 | 3 | >10 |
| RoBERTa | 61.7 | 70.9 | 68.6 | 67.6 | 71.0 | 71.1 | 73.1 | 74.5 |
| QAGNN | 65.1 | 74.5 | 74.2 | 72.1 | 71.6 | 75.6 | 71.3 | 78.6 |
| GreaseLM | 65.1 | 74.9 | 76.6 | 75.6 | 73.8 | 74.7 | 73.6 | 79.4 |
| DRAGON (**Ours**) | **75.2** | **79.6** | **77.5** | **79.1** | **78.2** | **77.8** | **80.9** | **83.5** |

Table 2: Accuracy of DRAGON on *CSQA + OBQA* dev sets for **questions involving complex reasoning** such as negation terms, conjunction terms, hedge terms, prepositional phrases, and more entity mentions. DRAGON consistently outperforms the existing LM (RoBERTa) and KG-augmented QA models (QAGNN, GreaseLM) in these complex reasoning settings.

tasks, but do not *pretrain* with a KG. GreaseLM is the existing top-performing model in this paradigm. As we use the same encoder architecture as GreaseLM for DRAGON, the only difference from GreaseLM is that DRAGON performs self-supervised pretraining while GreaseLM does not.

## 3.4 Results

Table 1 shows performance on the 9 downstream commonsense reasoning tasks. Across all tasks, DRAGON consistently outperforms the existing LM (RoBERTa) and KG-augmented QA models (QAGNN, GreaseLM), e.g., +7% absolute accuracy boost over RoBERTa and +5% over GreaseLM on *OBQA*. These accuracy boosts indicate the advantage of DRAGON over RoBERTa (KG reasoning) and over GreaseLM (pretraining). The gain is especially significant on datasets that have small training data such as *ARC*, *Riddle* and *OBQA*, and datasets that require complex reasoning such as *CosmosQA* and *HellaSwag*, which we analyze in more detail in the following sections.

### 3.4.1 Analysis: Effect of knowledge graph

The first key contribution of DRAGON (w.r.t. existing LM pretraining methods) is that we incorporate KGs. We find that this significantly improves the model's performance for robust and complex reasoning, such as resolving multi-step reasoning and negation, as we discuss below.

**Quantitative analysis.** In Table 2, we study downstream task performance of DRAGON on questions involving complex reasoning. Building on [8, 9], we consider several proxies to categorize complex questions: (i) presence of negation (e.g. *no*, *never*), (ii) presence of conjunction (e.g. *and*, *but*), (iii) presence of hedge (e.g. *sometimes*, *maybe*), (iv) number of prepositional phrases, and (v) number of entity mentions. Having negation or conjunction indicates logical multi-step reasoning, having more prepositional phrases or entity mentions indicates involving more reasoning steps or constraints, and having hedge terms indicates involving complex textual nuance. DRAGON significantly outperforms the baseline LM (RoBERTa) across all these categories (e.g., +14% accuracy for negation), which confirms that our joint language-knowledge pretraining boosts reasoning performance. DRAGON also consistently outperforms the existing KG-augmented QA models (QAGNN, GreaseLM). We find that QAGNN and GreaseLM only improve moderately on RoBERTa for some categories like conjunction or many prepositional phrases (=2, 3), but DRAGON provides substantial boosts. This suggests that through self-supervised pretraining with larger and diverse data, DRAGON has learned more general-purpose reasoning abilities than the finetuning-only models like GreaseLM.

**Qualitative analysis.** Using the *CSQA* dataset, we further conducted case studies on the behavior of DRAGON's KG reasoning component, where we visualize how graph attention weights change given different question variations (Figure 2). We find that DRAGON exhibits abilities to extrapolate and perform robust reasoning. For instance, DRAGON adjusts the entity attention weights and final predictions accordingly when we add conjunction or negation about entities (A1, A2) or when we add extra context to an original question (B1→B2), but existing models, RoBERTa and GreaseLM, struggle to predict the correct answers. As these questions are more complex than ones typically seen in the *CSQA* training set, our insight is that while vanilla LMs (RoBERTa) and finetuning (GreaseLM)

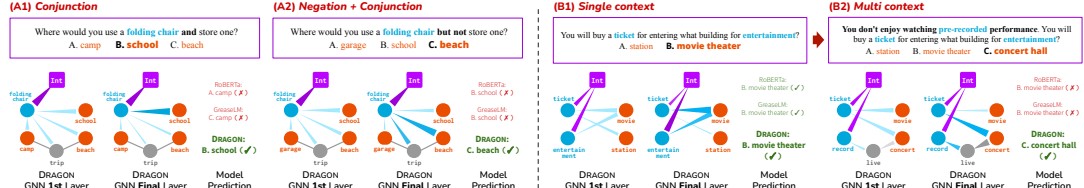

Figure 2: Analysis of DRAGON's graph reasoning, where we visualize how graph attention weights and final predictions change given question variations. Darker and thicker edges indicate higher attention weights. **DRAGON exhibits abilities to extrapolate and perform robust reasoning**. DRAGON adjusts the entity attention weights and final predictions accordingly when conjunction or negation is given about entities (A1, A2) or when extra context is added to an original question (B1→B2), but existing models, RoBERTa and GreaseLM, struggle to predict the correct answers. **A1:** DRAGON's final GNN layer shows strong attention to "school" but weak attention to "trip", likely because the question states "*and store one*"—hence, the chair is *not* used for a trip. **A2:** DRAGON shows strong attention to "trip" and "beach", likely because the question now states "*but not store one*"—hence, the chair *is* used for a trip. **B1→B2:** DRAGON's final GNN layer shows strong attention to "movie" in the original question (B1), but after adding the extra context "don't enjoy pre-record" (B2), DRAGON shows strong attention to "live" and "concert", leading to making the correctly adjusted prediction "concert hall". One interpretation of these findings is that DRAGON leverages the KG's graph structure as a scaffold for performing complex reasoning. This insight is related to recent works that provide LMs with scratch space for intermediate reasoning [8, 65, 66].

| Method | CosmosQA (10% train) | PIQA (10% train) |
|---|---|---|
| RoBERTa | 72.2 | 66.4 |
| GreaseLM | 73.0 | 67.0 |
| DRAGON (**Ours**) | **77.9** | **72.3** |

Table 3: Performance in low-resource setting where 10% of finetuning data is used. DRAGON attains large gains, suggesting its benefit for downstream data efficiency.

| Method | CSQA | OBQA |
|---|---|---|
| GreaseLM | 74.2 | 66.9 |
| GreaseLM-Ex | 73.9 | 66.2 |
| DRAGON (**Ours**) | 76.0 | 72.0 |
| DRAGON-Ex (**Ours**) | **76.3** | **72.8** |

Table 4: Downstream performance when model capacity—number of text-KG fusion layers—is increased ("-Ex"). Increased capacity does not help for the finetuning-only model (GreaseLM), but helps when pretrained (DRAGON), suggesting the promise of DRAGON to be further scaled up.

| Ablation Type | Ablation | CSQA | OBQA |
|---|---|---|---|
| Pretraining objective | MLM + LinkPred (**final**) | **76.0** | **72.0** |
| | MLM only | 74.3 | 67.2 |
| | LinkPred only | 73.8 | 66.4 |
| LinkPred head | DistMult (**final**) | **76.0** | **72.0** |
| | TransE | 75.7 | 71.4 |
| | RotatE | 75.8 | 71.7 |
| Cross-modal model | Bidirectional interaction (**final**) | **76.0** | **72.0** |
| | Concatenate at end | 74.5 | 68.0 |
| KG structure | Use graph (**final**) | **76.0** | **72.0** |
| | Convert to sentence | 74.7 | 70.1 |

Table 5: Ablation study of DRAGON. Using joint pretraining objective MLM + LinkPred (§2.3) outperforms using one of them only. All variants of LinkPred scoring models (DistMult, TransE, RotatE) outperform the baseline without LinkPred ("MLM only"), suggesting that DRAGON can be combined with various KG representation learning models. Cross-modal model with bidirectional modality interaction (§2.2) outperforms combining text and KG representations only at the end. Finally, using KG as graph outperforms converting KG as sentences, suggesting the benefit of graph structure for reasoning.

have limitation in learning complex reasoning, KG-augmented pretraining (DRAGON) helps acquire generalizable reasoning abilities that extrapolate to harder test examples.

### 3.4.2 Analysis: Effect of pretraining

Another key contribution of DRAGON (w.r.t. existing QA models like GreaseLM) is pretraining. Here we discuss when and why our pretraining is useful. Considering the three core factors in machine learning (data, task complexity, and model capacity), pretraining helps when the available downstream task data is smaller compared to the downstream task complexity or model capacity. Concretely, we find that DRAGON is especially helpful for the following three scenarios.

**Downstream tasks with limited data.** In Table 1, we find that DRAGON provides significant boosts over GreaseLM on downstream tasks with limited finetuning data available, such as *ARC* (3K training instances; +4% accuracy gain), *Riddle* (3K instances; +4% accuracy) and *OBQA* (5K instances; +5% accuracy). For other tasks, we also experimented with a low-resource setting where 10% of finetuning data is used (Table 3). Here we also see that DRAGON attains significant gains over GreaseLM (+5% accuracy on *PIQA*), suggesting the improved data-efficiency of DRAGON.

**Complex downstream tasks.** In Table 1, we find that DRAGON provides substantial gains over GreaseLM on downstream tasks involving more complex reasoning, such as *CosmosQA* and *HellaSwag*, where the inputs have longer context and more entities (thus bigger local KGs). For these tasks, improvements of GreaesLM over RoBERTa were small (+0.1% on *CosmosQA*), but DRAGON provides substantial boosts (+1.8%). Our insight is that through self-supervised pretraining

with larger and more diverse data, DRAGON has learned richer text-KG interactions than GreaseLM, enabling solving more complex downstream tasks. Similarly, as seen in §3.4.1, DRAGON also attains large gains over GreaseLM on complex questions containing negation, conjunction and prepositional phrases (Table 2), and extrapolates to questions more complex than seen in training sets (Figure 2).

**Increased model capacity.** In Table 4, we study downstream performance when the model capacity is increased—the number of text-KG fusion layers is increased from 5 to 7—for both GreaseLM and DRAGON. We find that increased capacity does not help for the finetuning-only model (GreaseLM) as was also reported in the original GreaseLM paper, but it helps when pretrained (DRAGON). This result reveals that increased model capacity can actually be beneficial when combined with pretraining, and suggests the promise of DRAGON to be further scaled up.

### 3.4.3 Analysis: Design choices of DRAGON

**Pretraining objective** (Table 5 top). The first important design choice of DRAGON is the joint pretraining objective: MLM + LinkPred (§2.3). Using the joint objective outperforms using MLM or LinkPred alone (+5% accuracy on *OBQA*). This suggests that having the bidirectional self-supervised tasks on text and KG facilitates the model to fuse the two modalities for reasoning.

**Link prediction head choice** (Table 5 middle 1). KG representation learning is an active area of research, and various KG triplet scoring models are proposed (Equation 9). We hence experimented with using different scoring models for DRAGON's link prediction head (§2.3). We find that while DistMult has a slight edge, all variants we tried (DistMult, TransE, RotatE) are effective, outperforming the baseline without LinkPred ("MLM only"). This result suggests the generality of DRAGON and its promise to be combined with various KG representation learning techniques.

**Cross-modal model** (Table 5 middle 2). Another core component of DRAGON is the cross-modal encoder with bidirectional text-KG fusion layers (§2.2). We find that if we ablate them and simply concatenate text and KG representations at the end, the performance drops substantially. This result suggests that deep bidirectional fusion is crucial to model interactions over text and KG for reasoning.

**KG structure** (Table 5 bottom). The final key design of DRAGON is that we leverage the graph structure of KGs via a sequence-graph encoder and link prediction objective. Here we experimented with an alternative pretraining method that drops the graph structure: we convert triplets in the local KG into sentences using a template [33], append them to the main text input, and perform vanilla MLM pretraining. We find that DRAGON substantially outperforms this variant (+2% accuracy on *OBQA*), which suggests that the graph structure of KGs helps the model perform reasoning.

## 4 Experiments: Biomedical domain

Biomedicine is a domain with extensive background knowledge [67, 68, 69, 1], and experts curate various knowledge bases for it [70, 17, 71, 72]. We hypothesize that these biomedical KGs can enable deeper understanding and reasoning about biomedical text. With this motivation, we pretrain DRAGON on a biomedical corpus and KG, and evaluate on biomedical downstream tasks.

**Pretraining setup.** For the text data, we use PubMed [73], a widely-used corpus in biomedial LM training (e.g., BioBERT [74], PubmedBERT [75]). It contains the abstracts of biomedical papers on PubMed and has 21GB of text. For the KG data, we use the Unified Medical Language System (UMLS) [17], a widely-used knowledge graph in biomedicine. It has 300K nodes and 1M edges in total. For training, we follow the same procedure as the experiment in the general domain (§3.1), except that we initialize DRAGON's LM component with BioLinkBERT-Large [19], the state-of-the-art biomedical LM, instead of RoBERTa-Large. Note that while "BioLinkBERT" has "Link" in its name, it is not about KG links but about citation links that the model was originally pretrained with.

**Downstream evaluation tasks.** We finetune and evaluate DRAGON on three popular biomedical NLP and reasoning benchmarks: MedQA-USMLE (**MedQA**) [76], **PubMedQA** [77], and **BioASQ** [78]. Appendix B.4 provides details on these tasks and data splits.

**Baselines.** We compare DRAGON with the vanilla LM (BioLinkBERT) and LMs finetuned with the KG (QAGNN and GreaseLM seeded with BioLinkBERT).

| Method | MedQA | PubMedQA | BioASQ |
|---|---|---|---|
| BioBERT [74] | 36.7 | 60.2 | 84.1 |
| PubmedBERT [75] | 38.1 | 55.8 | 87.5 |
| BioLinkBERT [19] | 44.6 | 72.2 | 94.8 |
| + QAGNN | 45.0 | 72.1 | 95.0 |
| + GreaseLM | 45.1 | 72.4 | 94.9 |
| DRAGON (**Ours**) | **47.5** | **73.4** | **96.4** |

Table 6: Accuracy on biomedical NLP tasks. DRAGON outperforms all previous biomedical LMs.

**Results.** Table 6 summarizes model performance on the downstream tasks. Across tasks, DRAGON outperforms all the existing biomedical LMs and KG-augmented QA models, e.g., +3% absolute accuracy boost over BioLinkBERT and +2% over GreaseLM on *MedQA*, achieving new state-of-the-art performance on these tasks. This result suggests significant efficacy of KG-augmented pretraining for improving biomedical reasoning tasks. Combined with the results in the general commonsense domain (§3.4), our experiments also suggest the domain-generality of DRAGON, serving as an effective pretraining method across domains with different combinations of text, KGs and seed LMs.

## 5 Conclusion

We presented DRAGON, a self-supervised pretraining method to learn a deeply bidirectional language-knowledge model from text and knowledge graphs (KGs) at scale. In both general and biomedical domains, DRAGON outperforms existing language models and KG-augmented models on various NLP tasks, and exhibits strong performance on complex reasoning such as answering questions involving long context or multi-step reasoning.

One limitation of DRAGON is that it is currently an encoder model (analogous to BERT) and does not perform language generation. An important future research would be to extend DRAGON to generation, and advance KG-enhanced language generation [28, 79].

## Reproducibility

Pretrained models, code and data are available at https://github.com/michiyasunaga/dragon. Experiments are available at
https://worksheets.codalab.org/worksheets/0xcf9cddffff864fb382e1a2f1393c8934.

## Acknowledgment

We thank Rok Sosic, Hamed Nilforoshan, Michael Moor, Qian Huang, members of the Stanford SNAP, P-Lambda, and NLP groups, as well as our anonymous reviewers for valuable feedback. We also gratefully acknowledge the support of HAI Google Cloud Credits 1051203844499; DARPA under Nos. HR00112190039 (TAMI), N660011924033 (MCS); ARO under Nos. W911NF-16-1-0342 (MURI), W911NF-16-1-0171 (DURIP); NSF under Nos. OAC-1835598 (CINES), OAC-1934578 (HDR), CCF-1918940 (Expeditions), IIS-2030477 (RAPID), NIH under No. R56LM013365; Stanford Data Science Initiative, Wu Tsai Neurosciences Institute, Chan Zuckerberg Biohub, Amazon, JPMorgan Chase, Docomo, Hitachi, Intel, JD.com, KDDI, Toshiba, NEC, and UnitedHealth Group. The content is solely the responsibility of the authors and does not necessarily represent the official views of the funding entities.

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
