# A Ethics, limitations and risks

We outline potential ethical issues with our work below. First, DRAGON is a method to fuse language representations and knowledge graph representations for joint reasoning. Consequently, DRAGON could reflect the same biases and toxic behaviors exhibited by language models and knowledge graphs that are used to initialize it. For example, language models have been shown to encode biases about race, gender, and other demographic attributes [80, 81] and generate toxic outputs [82]. Because DRAGON is seeded with pretrained language models that often learn these patterns, it is possible to reflect them in open-world settings. Second, the ConceptNet knowledge graph [7] used in this work has been shown to encode stereotypes [83], rather than completely clean commonsense knowledge. If DRAGON were used outside these standard benchmarks in conjunction with ConceptNet as a KG, it might rely on unethical relationships in its knowledge resource to arrive at conclusions. Consequently, while DRAGON could be used for applications outside these standard benchmarks, we would encourage implementers to use the same precautions they would apply to other language models and methods that use noisy knowledge sources.

Another source of ethical concern is the use of the MedQA-USMLE evaluation. While we find this clinical reasoning task to be an interesting testbed for DRAGON and for joint language and knowledge reasoning in general, we do not encourage users to use these models for real world clinical prediction.

Reference: [9].

# B Experimental Setup Details

## B.1 KG retrieval

Given each input text segment $W$, we follow the procedure from Yasunaga et al. [8] to retrieve a relevant local KG $G$ from the raw KG $\mathcal{G} = (\mathcal{V}, \mathcal{E})$. First, we use the entity linker from the spaCy library[1] to link entity mentions in $W$ to entity nodes in $\mathcal{G}$, obtaining an initial set of nodes $V_{el}$. Second, we add any bridge entities in $\mathcal{G}$ that are in a 2-hop path between any pair of linked entities in $V_{el}$ to get the total retrieved nodes $V \subseteq \mathcal{V}$. If the number of nodes in $V$ exceeds 200, we prune $V$ by randomly sampling 200 nodes from it to be the final retrieved nodes $V$. Lastly, we retrieve all the edges in $\mathcal{G}$ that connect any two nodes in $V$ to obtain $E \subseteq \mathcal{E}$, forming the final local KG, $G = (V, E)$.

## B.2 Graph initialization

For the ConceptNet knowledge graph used in the general commonsense domain (§3), we follow the method of MHGRN [33] to prepare the initial KG node embeddings. Specifically, we convert triplets in the KG into sentences using pre-defined templates for each relation. Then, these sentences are fed into BERT-Large [2] to compute embeddings for each sentence. Finally, for each entity, we collect all sentences containing the entity, extract all token representations of the entity's mention spans in these sentences, and return the mean pooling of these representations.

For the UMLS knowledge graph used in the biomedical domain (§4), node embeddings are initialized similarly using the pooled token output embeddings of the entity name from BioLinkBERT [19].

While extremely rare ($< 1\%$), in case when the input text does not yield any linked entity, we represent the graph using a dummy node initialized with 0, i.e., DRAGON backs off to only using the text side representations because the graph propagates no information.

## B.3 Hyperparameters

---

[1] https://spacy.io/

| Category | Hyperparameter | Commonsense domain | | Biomedical domain | |
|---|---|---|---|---|---|
| | | **Pretrain** | **Finetune** | **Pretrain** | **Finetune** |
| Model architecture | Number of text-KG fusion layers $M$ | 5 | 5 | 5 | 5 |
| | Number of Unimodal LM layers $N$ | 19 | 19 | 19 | 19 |
| | Number of attention heads in GNN | 2 | 2 | 2 | 2 |
| | Dimension of node embeddings and the messages in GNN | 200 | 200 | 200 | 200 |
| | Dimension of MLP hidden layers (except MInt operator) | 200 | 200 | 200 | 200 |
| | Number of hidden layers of MLPs | 1 | 1 | 1 | 1 |
| | Dimension of MInt operator hidden layer | 400 | 400 | 400 | 400 |
| Regularization | Dropout rate of the embedding layer, GNN layers and dense layers | 0.2 | 0.2 | 0.2 | 0.2 |
| Optimization | Learning rate of parameters in LM | 2e-5 | {1e-5, 2e-5, 3e-5} | 2e-5 | {1e-5, 2e-5, 3e-5} |
| | Learning rate of parameters not in LM | 3e-4 | {3e-4, 1e-3} | 3e-4 | {1e-4, 3e-4} |
| | Number of epochs in which LM's parameters are kept frozen | 2 | 4 | 2 | 4 |
| | Optimizer | RAdam | RAdam | RAdam | RAdam |
| | Learning rate schedule | linear warmup and decay | linear warmup and decay | linear warmup and decay | linear warmup and decay |
| | Warmup ratio | 0.1 | 0.1 | 0.1 | 0.1 |
| | Batch size | 8,192 | 128 | 8,192 | 128 |
| | Number of epochs | - | 10–70 | - | 10–70 |
| | Number of steps | 20,000 | - | 20,000 | - |
| | Max gradient norm (gradient clipping) | 1.0 | 1.0 | 1.0 | 1.0 |
| Data | Max number of nodes | 200 | 200 | 200 | 200 |
| | Max number of tokens | 512 | {128, 256} | 512 | 512 |

Table 7: Hyperparameter settings for models and experiments

## B.4 Downstream evaluation tasks

We use the following nine commonsense reasoning benchmarks for the experiments in the general domain (§3).

**CommonsenseQA (CSQA)** [56] is a 5-way multiple-choice QA task testing commonsense reasoning. The dataset has 12,102 questions. We use the in-house data splits by [32].

**OpenbookQA (OBQA)** [57] is a 4-way multiple-choice QA task containing elementary science questions. It has 5,957 questions. We use the original data splits in [36].

**RiddleSense (Riddle)** [58] is a 5-way multiple-choice task testing complex riddle-style commonsense reasoning. It has 5,715 questions. We split the dev set in half to make in-house dev/test sets.

**AI2 Reasoning Challenge, Challenge Set (ARC)** [59] is a 4-way multiple-choice QA task containing science exam questions. It has 2,590 questions. We use the original data splits in [59].

**CosmosQA** [60] is a 4-way multiple-choice QA task testing commonsense reasoning with long narratives. It has 35.6K questions. We split the dev set in half to make in-house dev/test sets.

**HellaSwag** [61] is a 4-way multiple-choice task testing grounded commonsense reasoning about events. It has 70K questions. We split the dev set in half to make in-house dev/test sets.

**Physical Interaction QA (PIQA)** [62] is a 3-way multiple-choice QA task testing physics reasoning about objects. It has 20K questions. We split the dev set in half to make in-house dev/test sets.

**Social Interaction QA (SIQA)** [63] is a 3-way multiple-choice QA task testing social commonsense reasoning. It has 37K questions. We use the original data splits in [63].

**Abductive Natural Language Inference (aNLI)** [64] is a 2-way multiple-choice task testing abductive commonsense reasoning. It has 170K questions. We use the original data splits in [64].

For the experiments in the biomedical domain (§4), we use the following three biomedical NLP and reasoning benchmarks.

**MedQA-USMLE (MedQA)** [76] is a 4-way multiple-choice task containing United States Medical License Exam questions. The dataset has 12,723 questions. We use the original data splits in [76].

**PubMedQA** [77] is a 3-way multiple-choice task testing biomedical language understanding and reasoning. The dataset has 1,000 questions. We use the original data splits in [77].

**BioASQ** [78] is a 2-way multiple-choice task testing biomedical language understanding and reasoning. The dataset has 885 questions. We use the original data splits in [78].

| Dataset | Example |
|---|---|
| CommonsenseQA | A weasel has a thin body and short legs to easier burrow after prey in a what? 
 (A) tree (B) mulberry bush (C) chicken coop (D) viking ship **(E) rabbit warren** |
| OpenbookQA | Which of these would let the most heat travel through? 
 (A) a new pair of jeans      **(B) a steel spoon in a cafeteria** 
 (C) a cotton candy at a store   (D) a calvin klein cotton hat |
| RiddleSense | What home entertainment equipment requires cable? 
 (A) radio shack (B) substation (C) cabinet **(D) television** (E) desk |
| AI2 Reasoning Challenge | Which property of a mineral can be determined just by looking at it? 
 **(A) luster** (B) mass (C) weight (D) hardness |
| CosmosQA | It's a very humbling experience when you need someone to dress you every morning, tie your shoes, and put your hair up. Every menial task takes an unprecedented amount of effort. It made me appreciate Dan even more. But anyway I shan't dwell on this (I'm not dying after all) and not let it detract from my lovely 5 days with my friends visiting from Jersey. What's a possible reason the writer needed someone to dress him every morning? 
 (A) The writer doesn't like putting effort into these tasks. **(B) The writer has a physical disability.** 
 (C) The writer is bad at doing his own hair.        (D) None of the above choices. |
| HellaSwag | A woman is outside with a bucket and a dog. The dog is running around trying to avoid a bath. She 
 (A) rinses the bucket off with soap and blow dries the dog's head. 
 (B) uses a hose to keep it from getting soapy. 
 **(C) gets the dog wet, then it runs away again.** 
 (D) gets into the bath tub with the dog. |
| Physical Interaction QA | You need to break a window. Which object would you rather use? 
 **(A) a metal stool** (B) a giant bear (C) a bottle of water |
| Social Interaction QA | In the school play, Robin played a hero in the struggle to the death with the angry villain. 
 How would others feel as a result? 
 (A) sorry for the villain **(B) hopeful that Robin will succeed** (C) like Robin should lose the fight |
| aNLI | Obs1: It was a gorgeous day outside. 
 Obs2: She asked her neighbor for a jump-start. 
 **Hyp1: Mary decided to drive to the beach, but her car would not start due to a dead battery.** 
 Hyp2: It made a weird sound upon starting. |
| MedQA-USMLE | A 57-year-old man presents to his primary care physician with a 2-month history of right upper and lower extremity weakness. He noticed the weakness when he started falling far more frequently while running errands. Since then, he has had increasing difficulty with walking and lifting objects. His past medical history is significant only for well-controlled hypertension, but he says that some members of his family have had musculoskeletal problems. His right upper extremity shows forearm atrophy and depressed reflexes while his right lower extremity is hypertonic with a positive Babinski sign. Which of the following is most likely associated with the cause of this patients symptoms? 
 (A) HLA-B8 haplotype     (B) HLA-DR2 haplotype 
 **(C) Mutation in SOD1**     (D) Mutation in SMN1 |
| PubMedQA | Recent studies have demonstrated that statins have pleiotropic effects, including anti-inflammatory effects and atrial fibrillation (AF) preventive effects [...] 
 221 patients underwent CABG in our hospital from 2004 to 2007. 14 patients with preoperative AF and 4 patients with concomitant valve surgery [...] 
 The overall incidence of postoperative AF was 26%. Postoperative AF was significantly lower in the Statin group compared with the Non-statin group (16% versus 33%, p=0.005). Multivariate analysis demonstrated that independent predictors of AF [...] 
 Do preoperative statins reduce atrial fibrillation after coronary artery bypass grafting? 
 **(A) yes** (B) no (C) maybe |
| BioASQ | LT4 absorption is unchanged by concomitant metformin ingestion. It has been hypothesized that metformin may suppress serum thyrotropin (TSH) concentrations by enhancing LT4 absorption or by directly affecting the hypothalamic-pituitary axis. Does metformin interfere thyroxine absorption? 
 (A) yes **(B) no** |

Table 8: Example for each downstream task dataset used in this work.

# C   Additional Experimental Results

| Method | Hit@3 |
|---|---|
| DistMult (i.e., KG only) | 61.3 |
| DRAGON (i.e., KG + text) | **78.1** |

Table 9: **KG link prediction performance on ConceptNet**. In addition to the NLP tasks we mainly used for downstream evaluation, DRAGON can also perform KG link prediction tasks in downstream. We find that DRAGON (which uses retrieved text besides the KG) achieves improved performance on the KG link prediction task compared to the baseline DistMult model (which does not use text).