# OpenReview forum: "Deep Bidirectional Language-Knowledge Graph Pretraining"
_NeurIPS.cc/2022/Conference — NeurIPS 2022 Accept_

### Official Review · Reviewer_YcoB · 2022-07-03

**Rating:** 5
**Confidence:** 3
**Soundness:** 3 good
**Presentation:** 4 excellent
**Contribution:** 2 fair

**Summary:**

This paper presents a method for jointly pre-training a BERT-style LM with a knowledge-graph component.
The paper builds on an existing architecture called GreaseLM, which can fuse information across a piece of text and a knowledge-(sub)graph using cross-attention. The contribution is that they extent Grease-LM by using pretraining, and demonstrate stronger results on a set of downstream tasks.

This is achieved by taking a chunk of text to be used in masked-language modelling, and running entity linking, and constructing a small knowledge graph from ConceptNet (or UMLS in the case of biomedical text).
The model is jointly to trained on masked-language modelling for the text, and link prediction for the KB subgraph.

The authors go on to finetune and test their model on a suite of multiple-choice commonsense-like QA tasks, and observe consistent improvements over a non-KG-augmented Baseline, and GREASE-LM (i.e. the non-pretrained variant of their model). They also show similar results on Biomedical tasks.

They finally present detailed ablations, demonstrate wider performance gaps for smaller training sets, and present some analysis.

**Questions:**

What happens when you use more (or fewer) nodes in the retrieved KG at fine-tune/test time? If I want a bigger neighbourhood (and slower inference), will my accuracy improve?

CommonsenseQA is constructed using data from ConceptNET, so the results you achieve here may be unfair compared to a text-only model. (I don't know if this is the case for the other datasets, but please address it in writing)


**Limitations:**

No specific concerns from me here. The may be a little bit of over-claiming in terms of the strength of the results.


**Strengths And Weaknesses:**

Strengths:
* Written clearly and easy to read.
* The authors experiment in two domains, (general English and Biomedical texts) and show consistent gains, strengthening their claims
* The authors demonstrate a good practice to treat the baseline Roberta-only model fairly by continuing to pretrain it for as long as the other model.
* the method works well compared to their baselines
* the ablations and analysis section are detailed and generous

Weaknesses/areas for improvement:
* There is a certain lack of novelty here, as the authors acknowledge, but the have done a good job of really trying to understand their design space well, and they ablations are thorough.
* The method requires an entity linker to work, which may not always be available or effective in the domain of interest.
* The construction of the pretraining data is probably one of the bigger factors in the success or failure of this model, I would like to see more details here in the main paper, rather than this be demoted to the appendix.
* less of a weakness, more of a suggestion: I think this would be significantly more impactful as a model/piece of work, if it was generative, and could generate text.
* If I've understood correctly, the method is static with respect to the knowledge graph (i.e. if the KG gets changed after pretraining time, the model will not have a good way to embed novel entities or relations) I think this is a missed opportunity, e.g. the nodes and relations are usually lexicalized, so the text-understanding part of the model could be used to create the node embeddings, so that the model could consume new entities and relations after pretraining.
* The quoted numbers for CSQA for the baseline look to be weak c.f. literature - a quick look at the leaderboard suggest Roberta achieves 72.1, whereas the authors quote 68.7, significantly lower. (https://www.tau-nlp.sites.tau.ac.il/csqa-leaderboard2) - Dragon is still better, but it makes me worry a little about the baseline numbers.
* Figure 2 is nice but really small, and pushes what is acceptable in my opinion.
* The gains in Table 4 gains seem very small - these should be accompanied by confidences/statistical significance tests.
* My main complaint is that the downstream experiments are quite narrow, limiting impact. Could this be used well on other kinds of NLP tasks? Further, The focus is entirely on what the model can do to improve NLP tasks using KGs, but I think it would be interesting to explore whether this model can be used to improve performance on KG tasks by using retrieved text - this would increase the scope and impact of this paper.
* commonsenseQA is constructed using data from ConceptNET, so the results you achieve here may be unfair compared to a text-only model. (I don't know if this is the case for the other datasets, but please address it in writing)

originality: Some lack of novelty, as mentioned above, but the authors have done enough to convince me this is a novel-enough contribution (combining existing things in new and interesting ways is a contribution)
quality: the paper is of good quality in my view - I appreciate the depth of ablation and analysis
clarity: very clear and easy to follow
significance: I have included above some areas which would improve significance and impact.

---

> ### Author Response · Authors · 2022-08-02
> **Author Response to Reviewer YcoB**
>
> We sincerely thank the reviewer for insightful feedback and for recognizing the novelty/depth of our work in studying the design space of text+KG models. We respond to the reviewer’s concerns and questions below (and in the general response above).
>
> > The method requires an entity linker to work, which may not always be available or effective in the domain of interest.
>
> For both the general and biomedical domains, **we actually used simple string matching as entity linker and found that this works well**. We think this is because different from existing methods that assume 1-1 alignment between entity mention in text and entity in KG, our method takes a KG *subgraph* as input, which can include multiple entity candidates (this can handle ambiguous entities) and their neighbors (this can handle synonyms). Our model will automatically learn what information from this KG subgraph to use in conjunction with the text input. Our intuition is that this helps the method be robust to noise in entity linking.
>
> > if the KG gets changed after pretraining time, the model will not have a good way to embed novel entities or relations
>
> This is a great question. **Our method can handle novel entities/relations**. To create input KG node embeddings, we lexicalize the entities and use the language model component to embed them (Appendix B.2). If there are new entities after pretraining, we follow the same step to prepare their node embeddings. Preparing the new node embeddings will not affect the embeddings of the existing nodes, because we freeze the node embeddings during pretraining (and instead use projection functions on these representations) so there's no distribution shift relative to the initialization. This is an advantage over traditional shallow KG embeddings (e.g. the original TransE). We will clarify this point in the final version.
>
> > The quoted numbers for CSQA for the baseline look to be weak c.f. the leaderboard…
>
> Thanks for pointing this out. Because the test set for CSQA is hidden, at the time of submission, we evaluated on the **in-house test split of CSQA, following the common practice in prior works**, e.g. KagNet, MHGRN, GreaseLM. Our RoBERTa/QAGNN scores in Table 1 match these prior works, e.g. GreaseLM Table 2 (https://arxiv.org/pdf/2201.08860.pdf). We will submit to the official CSQA leaderboard upon publication.
>
> > the downstream experiments are quite narrow, Could this be used well on other kinds of NLP tasks? … also it would be interesting to explore whether this model can be used to improve performance on KG tasks
>
> This is a great question. Because DRAGON inherits the structure of BERT (with KG added), **DRAGON can handle any task formats that BERT handles**, e.g. text/token classification, extractive QA, etc, besides multiple-choice tasks that we focused on in the submission. In fact, BioASQ and PubMedQA used in our biomedical downstream evaluation (Table 6) are general text classification tasks. We think it is worth mentioning that our current suite of downstream tasks (12 tasks; 9 in commonsense reasoning + 3 in biomedicine) is already broader than many existing text+KG works such as GreaseLM [ICLR’22] (3 tasks) and JAKET [AAAI’22] (3 tasks). Nevertheless, in the final version of our paper, we will further expand the scope and evaluate on more kinds of tasks such as extractive QA, GLUE and entity/relation classification.
>
> We also thank the reviewer for suggesting KG tasks. DRAGON can perform KG link prediction tasks and the table below is the result on ConceptNet KG. We find **DRAGON (which uses retrieved text besides KG) improves on the baseline KG link prediction model (DistMult without text)**. We will add this to the final version of our paper.
>
> **KG link prediction on ConceptNet**
> |Model|Hit@3|
> |-|-|
> |DistMult (i.e. KG only)|61.3|
> |DRAGON (i.e. KG + text)|**78.1**|
>
> > commonsenseQA is constructed using data from ConceptNET, so the results you achieve here may be unfair compared to a text-only model
>
> You are right, commonsenseQA is constructed using ConceptNet. Hence, **we also evaluated on various other datasets that did not use ConceptNet for construction**, such as OBQA, ARC, HellaSwag, CosmosQA, PIQA, SIQA, aNLI. We find that DRAGON achieves substantial performance gains (e.g. +7% on OBQA, +5% on ARC) on these tasks as well (Table 1).
>
> > What happens when you use more (or fewer) nodes in the retrieved KG at fine-tune/test time? If I want a bigger neighbourhood (and slower inference), will my accuracy improve?
>
> This is a very interesting question. DRAGON is able to handle bigger KG subgraphs at finetune/test time, but our current intuition is that using a KG subgraph of similar size to pretraining time tends to perform the best. It would be very interesting if we could pretrain with small KG subgraphs (making pretraining less expensive) but extrapolate to bigger KG subgraphs at inference time. This is a great future research - thank you for your suggestion.

---

> > ### Author Response · Authors · 2022-08-08
> > **Follow up on author response**
> >
> > Thank you again for taking your time to review our paper and providing detailed feedback. As the discussion period is ending shortly, we wanted to follow up to see if you may have any additional questions that we have not addressed. We have tried responding to all the comments from your initial review, and we are more than happy to have further discussion on any points. Thank you.

---

> > > ### Comment · Reviewer_YcoB · 2022-08-09
> > > **Thank you for response**
> > >
> > > Dear authors,
> > >
> > > Entity linker: whilst its true that the entity linker is not complex, it doesnt change the fundamental limitation that it is still required! but thank you for highlighting this anyway, it is certainly a benefit that a complex system is not needed for the domains you study.
> > >
> > > Thanks for explaining the entity/relation novelty procedure.
> > > It is good to hear that it is technically feasible to induce new items, but i would like to have seen this empirically validated ideally.
> > >
> > > > Thanks for pointing this out. Because the test set for CSQA is hidden, at the time of submission, we evaluated on the in-house test split of CSQA, following the common practice in prior works,
> > >
> > > This explanation makes sense, but the practice feels pretty unacceptable to me
> > >
> > > > DRAGON (which uses retrieved text besides KG) improves on the baseline KG link prediction model (DistMult without text)
> > >
> > > Cool!
> > >
> > > I have all that i need to be properly prepared going into the reviewer discussion. Thank you for your thorough rebuttal.

---

### Official Review · Reviewer_R7o8 · 2022-07-10

**Rating:** 6
**Confidence:** 4
**Soundness:** 3 good
**Presentation:** 3 good
**Contribution:** 3 good

**Summary:**

**Summary**

This work proposed a deep bidirectional language-knowledge pre-training, by leveraging both text and knowledge graphs. The proposed model has two core components: a cross-modal model that fuses text and KG bidirectionally, and a bidirectional self-supervised objective that learns joint reasoning over text and KG. The pre-training process contains two self-supervised reasoning tasks: masked language modeling (MLM), which masks and predicts tokens in the input text; and link prediction, which drops and predicts edges in the input KG. Experiments on two domains demonstrated the effectiveness of the proposed model.

**Questions:**

See weakness above.

**Ethics Review Area:**

["I don’t know"]

**Limitations:**

It seems the proposed method can only work on multi-choice QA problems. As shown in Figure 2, the proposed method needs to use answer choices to find relevant entity nodes on the knowledge graph. So, I am not sure how this method can be applied to other tasks, such as binary question answering tasks (e.g., CSQA2.0), and language generation tasks.

**Strengths And Weaknesses:**

**Strengths**

(1) The idea of jointly pre-training text and knowledge graph representations is well motivated. Most of the recent works focused on augmenting the model by fine-tuning it with a knowledge graph. Still, they were not pre-trained to learn deep fusion of the two modalities at scale, limiting the potential to acquire fully joint representations of text and knowledge graphs.

(2) The proposed method is able to take pairs of text and relevant knowledge graphs as input and bidirectionally fuses information from both. The experiments were conducted on two domains, both of which demonstrate superior performance to baseline methods.

**Weaknesses**

(1) The novelty of the proposed method is limited. First, jointly pre-training text and knowledge graph are not new, which has been studied in existing works [1, 2]. Second, the multi-modal MLM objective is widely used in vision-language pre-training works; the link prediction objective was first proposed by JAKET [2]. Third, the deep fusion between text and knowledge graph was proposed in GreaseLM [3].

(2) The experimental results of CSQA and OBQA were not consistent with existing works. Recent models have achieved 80+ accuracy on CSQA and OBQA, e.g., QA-GNN 76.1 on CSQA leaderboard, GreaseLM 84.8 on OBQA leaderboard. However, the performance in the paper is lower than the baseline methods by a large margin. Did the authors submit the results to the leaderboard?

(3) The title of this paper is confusing, especially the phrase “language-knowledge”. Why does language not include knowledge?

[1] JointGT: Graph-Text Joint Representation Learning for Text Generation from Knowledge Graphs. ACL 2021 (This is a missing reference.)

[2] Jaket: Joint pre-training of knowledge graph and language understanding. AAAI 2022

[3] Greaselm: Graph reasoning enhanced language models for question answering. ICLR 2022

---

> ### Author Response · Authors · 2022-08-02
> **Author Response to Reviewer R7o8**
>
> We sincerely thank the reviewer for constructive feedback and for describing our work as well-motivated and demonstrating superior performance. We respond to the reviewer’s concerns below and in the general response above.
>
> > (1) novelty of the proposed method is limited.
>
> Thanks for pointing to related works such as vision-language models, GreaseLM, JAKET, a new one, JointGT. We have now included JointGT in our paper. **We have responded to this concern in detail in the general response above**, and we would appreciate it if the reviewer could take a look. In short, 1) our work solves different and **orthogonal problems** than the mentioned prior works; 2) we show **novel results** where our self-supervised text+KG pretraining leads to significant improvements on various QA, reasoning and text classification tasks, beyond the classical entity/relation tasks that existing text+KG works tend to focus on; 3) our work presents **methodological novelties** where we answer various open questions and design choices in the space of learning from text+KG, and show a carefully-designed method recipe that produces a significantly more performant model.
>
> > (2) experimental results of CSQA and OBQA were not consistent with existing works and leaderboard.
>
> Thanks for pointing this out. **We have responded to this point in detail in the general response above.** In short, this is because the OBQA and CSQA dataset have multiple evaluation settings. The OBQA result in Table 1 is the setup where we do **not** use the extra science facts prepared by https://arxiv.org/abs/1909.01958. We also have our results when these extra science facts are used in **Appendix Table 9, where our DRAGON achieves 87.7, which strongly outperforms GreaseLM’s 84.8**. We apologize for the confusion and will move this result to the main body in the final version.
> Re: CSQA, because the test set for CSQA is hidden, at the time of submission, we evaluated on the in-house test split of CSQA, following the common practice in prior works, e.g. KagNet, MHGRN, GreaseLM. We will submit to the official CSQA leaderboard upon publication.
>
> > (3) The title of this paper is confusing, especially the phrase “language-knowledge”.
>
> Thanks for your valuable feedback on the title. We will make it clearer, e.g. “Deep Bidirectional Language-Knowledge Graph Pretraining” to clarify “knowledge” means “knowledge graphs”. Please let us know if you don’t think this is a suitable alternative.
>
> > It seems the proposed method can only work on multi-choice QA. how this method can be applied to other tasks, such as binary question answering (CSQA2.0), and language generation.
>
> This is a great question. Because DRAGON inherits the structure of BERT (with KG added), **DRAGON can handle any task formats that BERT handles**, e.g. general text/token classification besides multiple-choice tasks. When there are no answer choices, we can do entity linking simply for the main text to retrieve a KG subgraph. In fact, BioASQ and PubMedQA used in our biomedical downstream evaluation (Table 6) are binary and 3-way text classification tasks. We also experimented with **CSQA2.0**, and the result is the table below. We find a consistent trend that DRAGON improves on RoBERTa/GreaseLM.
>
> **DRAGON can also perform KG link prediction tasks** and the table below is the result on ConceptNet KG. We will add this to our paper.
>
> **CSQA2.0**
> |Model|IH-Test Acc.|
> |-|-|
> |RoBERTa-large|56.7|
> |GreaseLM|57.0|
> |DRAGON|**58.8**|
>
> **KG link prediction on ConceptNet**
> |Model|Hit@3|
> |-|-|
> |DistMult (i.e. KG only)|61.3|
> |DRAGON (i.e. KG + text)|**78.1**|
>
> As DRAGON inherits the format of BERT encoder, it itself does not do language generation, just like BERT. But DRAGON can be extended to help language generation: (1) DRAGON can initialize the encoder of seq2seq language model; (2) insights of DRAGON can also be applied to build GPT-style decoder language model by dynamically retrieving and fusing relevant KG subgraph on the fly as it generates next words. These are interesting followup research of DRAGON.
>
> We would like to thank the reviewer again for their time, and we are happy to have any further discussion.

---

> > ### Author Response · Authors · 2022-08-08
> > **Follow up on author response**
> >
> > Thank you again for taking your time to review our paper and providing detailed feedback. As the discussion period is ending shortly, we wanted to follow up to see if you may have any additional questions that we have not addressed. We have tried responding to all the comments from your initial review, and we are more than happy to have further discussion on any points. Thank you.

---

> > ### Comment · Reviewer_R7o8 · 2022-08-09
> > **Your reply addresses my concerns. I increased my rating!**
> >
> > Many thanks to the authors for the new experiments!
> >
> > -- I can now understand the performance difference (with and without retrieved facts) on the OBQA dataset. I recommend that the authors include these new results in the final version of the paper.
> >
> > -- New experiments on CSQA2.0 and KG link prediction are impressive and address my concerns.
> >
> > -- The new title is better than the previous one. The term "language-knowledge" is confusing.
> >
> > -- Also, DRAGON cannot be applied to NLG tasks, which should also be discussed in the paper. Some related KG-enhanced NLG papers should be cited [1] [2], or possibly adding the discussion of applying DRAGON to NLG area as future work.
> >
> > [1] Jointgt: Graphical-text joint representation learning for generating text from knowledge graphs. ACL 2021.
> > [2] A Survey of Knowledge Augmented Text Generation. ACM Computing Survey 2022.

---

> > > ### Author Response · Authors · 2022-08-09
> > > **Thank you!**
> > >
> > > We sincerely thank the reviewer for taking our response into consideration.
> > >
> > > We also greatly appreciate the suggestion on citing and discussing KG-enhanced NLG - we have now added them to our draft.

---

### Official Review · Reviewer_Vj1c · 2022-07-10

**Rating:** 8
**Confidence:** 4
**Soundness:** 4 excellent
**Presentation:** 4 excellent
**Contribution:** 4 excellent

**Summary:**

Authors propose DRAGON - a self-supervised method to pretrain a language-knowledge model from raw text and KQ at scale. A training instance is formulated from a text segment and the related sub-knowledge graph extracted based on the entities appearing in the text segment. The model bidirectionally fuses information from both modalities. The pre-training step is done on BookCorpus and ConceptNet and fine-tuning is performance on specific test-data. Different from GreaseLM where there is only the finetuning step, the pretraining step of DRAGON helps the finetuned model and shows noticeable gain in downstream QA datasets.

**Questions:**

1. In the KG retrieval for each text segment, it only considers matching with the entities but not the relationships between them. Can you discuss the impact if the edge's relationship doesn't match with the one described in a text segment?

2. Did authors try to evaluate the performance of model on knowledge graph embedding tasks?

**Limitations:**

Yes, authors have discussed about the ethics, limitations and risks in Appendix A

**Strengths And Weaknesses:**

Originality: Authors present an effective approach to learn deep fusion of text data and knowledge graph at scale, applied for pre-training language-knowledge model. The pretrained model has promising performance when being finetuned on downstream QA tasks.

Quality: The proposed idea is technically sound, and the experiment results are convincing. Authors also discuss different design choices of DRAGON, which provides insights and explain why the model works.

Clarity: The paper is well written and easy to follow. I like that authors use the main example in Fig 1 to explain the key idea of the designed components (MLM and LinkPred losses). Another comment is that it will be useful to show an example of text segment and its related sub-graph (can be in the Appendix if there is no space).

Significance: Given the novelty and effectiveness of the proposed idea, the contribution of this work is significant.

---

> ### Author Response · Authors · 2022-08-02
> **Author Response to Reviewer Vj1c**
>
> We sincerely thank the reviewer for describing our work as novel, technically sound and convincing. We respond to the reviewers’ questions below.
>
> > In the KG retrieval for each text segment, it only considers matching with the entities but not the relationships between them. Can you discuss the impact if the edge's relationship doesn't match with the one described in a text segment?
>
> This is a great question. When we retrieve a KG subgraph, we first match the mentioned entities and then take all edges that span these entities in the KG, which can include relations that are not explicitly mentioned in the text, as well as ones mentioned in the text. The idea is that our goal in using KG is not just to provide the same knowledge found in text, but also to augment with other relationships/entities that might be latent to the described situation. For example, in the text “Where would you find a basement that can be accessed with an elevator?”, the explicit entity mentions are “basement” and “elevator”, but by taking their neighbor entities and all edges spanning them in the KG, we can get useful relations and bridging entities that were not explicitly mentioned in the text, such as “<elevator, at_location, office_building>”, “<basement, is_part_of, office_building>”. These latent knowledge triplets may help reason about the original question and provide answers such as “office_building”. We think that provision of such latent background knowledge is one of the major benefits of combining KG with text.
>
> > Did authors try to evaluate the performance of model on knowledge graph embedding tasks?
>
> Yes. We have experimented with KG link prediction tasks, and the table below is the result on ConceptNet KG. We find that DRAGON (which uses retrieved text besides KG) improves on the corresponding baseline KG link prediction model (DistMult without text). We will add this to the final version of our paper. We thank the reviewer for the valuable suggestion.
>
> **KG link prediction on ConceptNet**
> |Model|Hit@3|
> |-|-|
> |DistMult (i.e. KG only)|61.3|
> |DRAGON (i.e. KG + text)|**78.1**|

---

> > ### Author Response · Authors · 2022-08-08
> > **Follow up on author response**
> >
> > Thank you again for taking your time to review our paper and providing detailed feedback. As the discussion period is ending shortly, we wanted to follow up to see if you might have any additional questions that we have not addressed. We have tried responding to the comments from your initial review, and we are more than happy to have further discussion on any points. Thank you.

---

### Author Response · Authors · 2022-08-02
**General Response to All Reviewers**

We sincerely thank our reviewers for their time and insightful feedback. We will incorporate all suggestions to our final version. We appreciate Reviewer Vj1c and YcoB’s positive review that our work provides novel insights into the design space of text+KG models and produces strong results.

Here we respond to common/main concerns raised by reviewers. We have also responded to reviewers individually.

> OBQA/CSQA results (R7o8, YcoB)

Reviewer R7o8, YcoB point out that our OBQA/CSQA results in Table 1 differ from the leaderboard websites.

This is because the OBQA dataset supports multiple evaluation settings. The OBQA result in Table 1 is the setup where we only use the question text as input and do **not** use the extra science facts prepared by https://arxiv.org/abs/1909.01958. This follows the practice in prior works (e.g. KagNet, MHGRN, QAGNN) and our RoBERTa/QAGNN scores match QAGNN Table 4 (https://arxiv.org/pdf/2104.06378.pdf). Systems submitted to the leaderboard typically use the extra science facts for better performance. In fact, we also have our results in this setup in **Appendix Table 9, where our DRAGON achieves 87.7, which strongly outperforms GreaseLM’s 84.8**. We will clarify and move the result to the main body in the final version.

Re: CSQA, because the test set for CSQA is hidden, at the time of submission, we evaluated on the **in-house test split of CSQA, following the common practice in prior works**, e.g. KagNet, MHGRN, GreaseLM. Our RoBERTa/QAGNN scores in Table 1 match these prior works, e.g. GreaseLM Table 2 (https://arxiv.org/pdf/2201.08860.pdf). We will submit to the official CSQA leaderboard upon publication.

> Novelty (R7o8)

Reviewer R7o8 is concerned about the novelty of our work with respect to works such as vision-language models, JointGT, GreaseLM, JAKET. We thank the reviewer for pointing to these related works, but our work focuses on different and **orthogonal problems**. Vision-language is a substantially different modality than the language-KG setting we focus on; JointGT does KG-to-text generation whereas we work on joint encoder for text+KG; GreaseLM does supervised finetuning whereas we propose self-supervised pretraining; JAKET solves entity/relation classification tasks whereas we focus on QA and reasoning tasks (which are arguably more generic).

In particular, we would like to emphasize the **novelty of our results**: while many existing pretrained text+KG models (e.g. JAKET, KnowBert) focus on solving entity/relation classification tasks in downstream (which KG augmentations would be well-suited for), we show that our self-supervised text+KG pretraining leads to significant improvements on various QA, reasoning and text classification tasks, where whether/how much pretraining with KG can help is less obvious. Our work provides a key insight that with a carefully-designed self-supervision method like what we propose, pretraining with KG can help broader NLP tasks beyond classical entity/relation tasks.

Moreover, as pointed out by Reviewer Vj1c and YcoB, our work has substantial **methodological novelties**. There are various open questions and important design choices in the space of jointly learning from text and KG that were not studied systematically. We provide answers to these open questions through extensive ablations:
- Do we need to pretrain with KG or can we just finetune? (DRAGON vs. GreaseLM) (Sec 3.4.2)
- Effect of bidirectional interaction over text and KG (Table 5, 4)
- Variation in self-supervised objectives (masked LM? link prediction? What KG objective: TransE, DistMult, RotatE?) (Table 5)
- Variation in KGs and domains (commonsense, biomedicine) (Sec 4)

Consequently, we present a carefully-designed method recipe that produces a significantly more performant model.

> Scope of downstream tasks (R7o8, YcoB)

Reviewers R7o8, YcoB ask if our DRAGON model can be used beyond the multiple-choice tasks we focused on. Because DRAGON inherits the structure of BERT (with KG added), **DRAGON can handle any task formats that BERT handles**, e.g. text classification, sequence labeling (not just multiple-choice tasks). In fact, BioASQ and PubMedQA used in our biomedical downstream evaluation (Table 6) are general text classification tasks. In response to Reviewer R7o8, we also experimented with CSQA2.0 (binary classification) and added the result below. We find a consistent trend that DRAGON improves on RoBERTa/GreaseLM.

**CSQA2.0**
|Model|IH-Test Acc.|
|-|-|
|RoBERTa-large|56.7|
|GreaseLM|57.0|
|DRAGON|**58.8**|

We thank Reviewer YcoB, Vj1c for suggesting KG tasks. **DRAGON can perform KG link prediction tasks** and the table below is the result on ConceptNet KG. We find DRAGON (which uses retrieved text besides KG) improves on the baseline KG link prediction model (DistMult without text). We will add this to our paper.

**KG link prediction on ConceptNet**
|Model|Hit@3|
|-|-|
|DistMult (i.e. KG only)|61.3|
|DRAGON (i.e. KG + text)|**78.1**|

---

### Meta-Review · Area_Chair_KTnM · 2022-08-25

**Recommendation:** Accept
**Confidence:** Less certain

**Metareview:**

This paper describes a pretraining approach that can leverage both text and knowledge graphs. The model has a cross-modal model that fuses text and KG bidirectionally, and a bidirectional self-supervised objective that learns joint reasoning over text and KG. For pretraiing the models uses traditional masked language modeling (MLM) as in BERT and link prediction, which drops and predicts edges in the input KG. Experiments on two domains demonstrate the effectiveness of the pretraining method.

Reviewers agree that the paper will make a good addition to NeurIPS, but disagree in their level of enthusiasm. The concerns expressed is that the approach is not particularly novel and that it requires entity linking, which adds more complexity to pretraining. Nevertheless, the reviewers agree that adding knowledge graph information to pretrained models is important, the experimental results are convincing, and the paper is clear and easy to read.

There was a productive discussion between the reviewers and the authors. As result, the paper was improved by adding results and clarifications. The improved version will make a good contribution to the conference program.

**Award:**

No

---

### Decision · Program_Chairs · 2022-09-14

Accept